# Effects of Microbial Fertilizer Combined with Organic Fertilizer on Forage Productivity and Soil Ecological Functions in Grasslands of the Muli Mining Area

**DOI:** 10.3390/plants14203156

**Published:** 2025-10-14

**Authors:** Zongcheng Cai, Jianjun Shi, Shouquan Fu, Fayi Li, Liangyu Lv, Qingqing Liu, Hairong Zhang, Shancun Bao

**Affiliations:** 1Academy of Animal Husbandry and Veterinary Sciences, Qinghai University, Xining 810016, China; ys230951310630@qhu.edu.cn (Z.C.); ys240951310609@qhu.edu.cn (S.F.); lfy99218@qhu.edu.com (F.L.); yb220909000082@qhu.edu.cn (Q.L.); 15500620398@163.com (H.Z.); bshancun@163.com (S.B.); 2Key Laboratory of Adaptive Management of Alpine Grassland, Xining 810016, China; 3State Key Laboratory of Ecology and Plateau Agriculture and Animal Husbandry in Sanjiangyuan Jointly Established by the Ministry of Provincial Affairs, Qinghai University, Xining 810016, China

**Keywords:** alpine mining area, effective microorganisms inoculant, productivity of artificial grassland, soil remediation, soil enzyme activity, microbial biomass

## Abstract

To address grassland ecosystem degradation caused by mining disturbance and its severe threats to regional ecological security in alpine mining areas, this study systematically evaluated the synergistic effects of different application ratios of Effective Microorganisms inoculant and organic fertilizers on artificial grassland ecosystem functions in the Muli alpine mining region of the Qinghai-Tibet Plateau, based on field experiments conducted from 2022 to 2024. The results demonstrated significant improvements in production performance. The Y2E2 treatment (0.60 t·hm^−2^ Effective Microorganisms inoculant + 20 t·hm^−2^ organic fertilizer) exhibited optimal effects, with aboveground biomass increasing by 75.97% and 68.88% in 2023 and 2024, respectively, compared to the control (*p* < 0.05), while belowground biomass simultaneously increased by 36.05% and 35.53% (*p* < 0.05), showing a sustained upward trend. Nutritional quality was markedly enhanced, with the Y2E2 treatment consistently achieving the best performance across both years. Crude protein and ether extract contents increased by 46.18%~46.52% and 62.42%~63.25%, respectively (*p* < 0.05), while soluble sugar content rose significantly by 19.49%~20.56% (*p* < 0.05). Concurrently, crude ash and fiber fractions were significantly reduced. Soil physicochemical properties improved substantially, with the Y2E2 treatment in 2024 reducing soil pH and bulk density by 11.10% and 37.20%, respectively (*p* < 0.05), while increasing soil organic carbon, available nitrogen, and available potassium by 92.94%, 49.25%, and 96.08% (*p* < 0.05). Soil biological activity was significantly enhanced, as evidenced by increases of 78.33%, 55.69%, 55.87%, and 183.67% in β-glucosidase, dehydrogenase, urease, and acid phosphatase activities, respectively (*p* < 0.05), alongside rises of 117.64% and 94.78% in microbial biomass carbon and phosphorus (*p* < 0.05). Mechanistic analysis via structural equation modeling revealed strong positive direct effects of the Effective Microorganisms inoculant–organic fertilizer combination on forage yield (*β* = 0.27, *p* < 0.001) and nutritional quality (*β* = 0.73, *p* < 0.001). Principal component analysis (cumulative variance explained: 88.90%) further confirmed Y2E2 treatment superior performance in soil improvement, microbial function enhancement, and grassland productivity. In conclusion, the optimal remediation strategy for alpine mining grasslands was identified as the combined application of 0.60 t·hm^−2^ Effective Microorganisms inoculant and 20 t·hm^−2^ organic fertilizer. This approach drives ecosystem function restoration through a multidimensional synergistic mechanism involving soil physicochemical amelioration–microbial activity stimulation–nutrient supply optimization, providing both theoretical foundations and practical solutions for ecological restoration of degraded grasslands in similar regions.

## 1. Introduction

Alpine mining ecosystems, functioning as both globally climate-sensitive zones and critical nodes in biogeochemical cycles, possess particular ecological significance in maintaining Earth’s cryosphere functions [1]. The Muli alpine mining area, situated on the northeastern margin (elevation >4000 m), serves as a core ecological barrier within the “Third Pole” region. The degradation of its grassland ecosystems has not only diminished regional grassland productivity and forage quality but also intensified soil erosion risks, thereby compromising the area’s ecological security barrier functions [2,3]. Conventional soil remediation technologies have primarily focused on physical or chemical amendments, yet they are constrained by limitations including prohibitive costs and substantial environmental disturbances [4]. In recent years, microbial-based ecological restoration strategies have garnered increasing attention due to their environmental compatibility and sustainability advantages. Among these, the synergistic application of Effective Microorganisms (EM) microbial inoculant with organic fertilizer demonstrated potential for multidimensional improvement in soil–plant system functions, offering novel approaches for ecological restoration in alpine mining areas [5].

Fertilization, as a core technical means for ecological restoration in mining areas, plays a crucial role in the reconstruction of artificial communities by improving vegetation productivity and soil functions [6]. Research has shown that reasonable fertilization can significantly increase vegetation coverage, biomass, and community diversity, optimize soil nutrient storage capacity and enzyme activity, and reconstruct the structure and functional diversity of microbial communities [7]. Compared with traditional fertilization, micro-ecological restoration technology has established a synergistic interaction system of microorganisms–organic matter–plants by introducing compound microbial inoculant and organic fertilizers (such as PGPR microbiota including nitrogen-fixing bacteria and phosphate-solubilizing bacteria). It shows unique advantages in improving the soil micro-environment, promoting nutrient cycling, and enhancing system sustainability [8,9].

Previous studies have demonstrated the efficacy of microbial inoculants in ecological restoration. Sun et al. [10] reported that EM microbial inoculant application increased crude protein and soluble sugar contents by 18.30% and 14.70%, respectively, in grass–legume mixed pastures, significantly improving forage nutritional quality. Tu et al. [11] further revealed that a microbial–organic fertilizer combination enhanced soil water content by 37.00%, increased porosity by 22.00%, and boosted available nitrogen and phosphorus contents by 40.00%~55.00% in arid mining areas. Through metagenomic analysis, Yan et al. [12] observed that microbial inoculants elevated the relative abundance of nitrogen-fixing bacteria (e.g., *Bradyrhizobium* spp.) and methanotrophs (*Methylocystis* spp.) by 2.3~4.5 fold in coal mine spoils, effectively restructuring soil microbial communities and enhancing carbon–nitrogen cycling processes. Yuan et al. [13] demonstrated that microbial inoculants accelerated organic carbon accumulation by 28.60% in bauxite mining reclamation areas, with stable humus fractions increasing by 12.40% within carbon components. Notably, Dang et al. [14] found potassium-solubilizing bacteria inoculation increased available potassium content by 68.00% while stimulating urease and sucrase activities by 52.00% and 41.00%, respectively, in mining soils. Zhang et al. [15] confirmed that arbuscular mycorrhizal fungi inoculation enhanced sea buckthorn biomass by 67.00% in coal mining subsidence areas, with mean weight diameter of soil aggregates increasing 0.35 mm, accompanied by 29.00% and 44.00% improvements in organic matter content and phosphatase activity, highlighting the synergistic advantages of mycorrhizal symbiosis systems. Complementary findings by Yin et al. [16] established that AMF significantly improved soil aggregate stability, enzyme activities, soil organic carbon content, and ryegrass biomass during short-term remediation periods. However, most of the existing studies focus on a single restoration goal (such as yield improvement or soil improvement). The effects of the combined technology of microbial inoculant and organic fertilizers on the yield and nutritional quality of forage in artificial grasslands—as well as the regulatory effects on both soil physicochemical properties and key biochemical indicators, enzyme activity, and microbial biomass carbon, nitrogen, and phosphorus—still need in-depth exploration.

Based on these findings, this study addresses the critical need for ecological restoration in the Muli alpine mining grassland by aiming to develop a synergistic remediation system that concurrently enhances vegetation productivity and restores soil ecological functions. We propose the following hypotheses: (1) The combined application of EM microbial inoculants and organic fertilizers will significantly improve both aboveground biomass and forage nutritional quality in artificial grasslands by ameliorating soil micro-ecological conditions; (2) Gradient application treatments will markedly optimize soil physicochemical properties and key enzyme activities through the regulation of soil biochemical processes; (3) The introduction of EM inoculants will effectively modulate the dynamics of microbial biomass carbon, nitrogen, and phosphorus, thereby enhancing microbial functional diversity and triggering cascading response mechanisms within the plant–soil system.

Through a gradient application experiment of EM inoculants and organic fertilizers, we systematically analyzed vegetation growth dynamics and soil quality evolution characteristics across different treatments. Principal component analysis was employed to identify the optimal application regime, while structural equation modeling was used to elucidate the hypothesized cascading response mechanisms. This study provides both theoretical foundations and practical guidance for ecological restoration of artificial grasslands in alpine mining regions.

## 2. Results

### 2.1. Effects of Combined Microbial Inoculant and Organic Fertilizer Application on Artificial Grassland Biomass

As shown in Figure 1, fertilization treatments significantly influenced both aboveground and belowground biomass production in the artificial grassland during the 2023–2024 study period. For aboveground biomass (Figure 1a,c), the control treatment (CK) consistently yielded the lowest values, demonstrating significant differences from all co-application treatments (*p* < 0.05). The Y2E2 treatment achieved the highest aboveground biomass, measuring 349.00 g·m^−2^ in 2023 (75.97% increase over CK) and 360.67 g·m^−2^ in 2024 (68.88% increase over CK), with both values being significantly superior to other treatments (*p* < 0.05).

Similarly, belowground biomass showed the lowest values under CK treatment (Figure 1b,d), recording only 413.33 g·m^−2^ in 2023 and 420.33 g·m^−2^ in 2024. In contrast, Y2E2 treatment maintained the highest belowground biomass, reaching 562.33 g·m^−2^ (36.05% increase over CK) in 2023 and 569.67 g·m^−2^ (35.53% increase over CK) in 2024, significantly exceeding all other treatments (*p* < 0.05). The increasing biomass trend from 2023 to 2024 confirmed the sustained positive effects of co-application fertilization.

These results demonstrated that optimized fertilization combinations significantly enhance both aboveground and belowground biomass accumulation in grassland vegetation, with distinct treatment-specific effects showing significant heterogeneity. Notably, the Y2E2 treatment exhibited the most pronounced growth-promoting effects.

### 2.2. Effects of Combined Application of Microbial Inoculants and Organic Fertilizers on Forage Nutritional Quality in Artificial Grassland

The 2023 data (Figure 2a,d) demonstrated that crude protein and ether extract contents reached the highest levels in the Y2E2 treatment, showing significant increases of 46.18% and 63.25%, respectively, compared to the CK (*p* < 0.05). Soluble sugar content ranged from 9.42 to 11.26 mg·g^−1^, with the Y2E3 treatment yielding the maximum value of 11.26 mg·g^−1^, representing a 19.49% significant increase over CK (*p* < 0.05). All combined fertilization treatments reduced both crude ash and fiber contents compared to CK. Specifically, the Y2E2 treatment showed the lowest ash (3.65%) and neutral detergent fiber (56.31%) contents, corresponding to significant reductions of 29.18% and 15.85% versus CK (*p* < 0.05). Except for Y1E1, all combined fertilization treatments maintained acid detergent fiber below 40.00%, with the most pronounced reduction in Y3E1 (30.36%, a 28.40% significant decrease from CK; *p* < 0.05).

The 2024 data confirmed that the Y2E2 treatment consistently optimized forage nutritional quality (Figure 2c,d), maintaining the highest contents of crude protein (12.86%), ether extract (4.34%), and soluble sugars (11.63 mg·g^−1^), representing significant increases of 46.52%, 62.42%, and 20.56%, respectively, compared to the CK treatment (*p* < 0.05). Crude ash content ranged from 3.53% to 4.98%, with the Y2E3 treatment showing the lowest value (29.00% significant reduction versus CK; *p* < 0.05). Both acid detergent fiber (30.26%) and neutral detergent fiber (54.52%) contents in the Y2E2 treatment were significantly lower than CK, demonstrating reductions of 27.87% and 17.01%, respectively (*p* < 0.05).

These results demonstrated that the Y2E2 fertilization treatment significantly improved the nutritional composition of artificial grassland forage by enhancing crude protein and ether extract contents while reducing ash and fiber contents, with consistent optimal performance across both years, thereby effectively improving the nutritional quality of artificial grassland forage in alpine mining areas.

### 2.3. Effects of Combined Application of Microbial Inoculants and Organic Fertilizers on the Physical and Chemical Properties of Artificial Grassland Soil

As demonstrated in Table 1, the combined fertilization treatments significantly improved soil physicochemical properties in artificial grasslands of alpine mining areas. Soil pH values ranged from 8.05 to 8.78, with the control (CK) treatment showing the highest values, while the combined fertilization treatments significantly reduced soil pH, reaching the lowest value in the Y2E2 treatment (8.31% significant reduction compared to CK, *p* < 0.05). Soil bulk density was highest in the CK treatment, and the combined fertilization treatments significantly reduced bulk density, with the Y3E2 treatment showing the lowest value (0.93 g·cm^−3^, representing a 46.55% significant reduction compared to CK, *p* < 0.05). The combined fertilization treatments enhanced soil available nutrient contents, with soil organic carbon, alkali-hydrolyzable nitrogen, and available potassium all reaching their highest levels in the Y2E2 treatment, showing significant increases of 72.72%, 50.98%, and 96.88%, respectively, compared to CK (*p* < 0.05). Available phosphorus content ranged from 19.70 to 39.23 mg·kg^−1^, being highest in the Y2E3 treatment, followed by Y2E2, while the CK treatment showed significantly lower content than all other combined fertilization treatments (*p* < 0.05).

Table 2 presents the sustained effects of combined fertilization treatments on soil physicochemical properties in the alpine mining area artificial grasslands during 2024. Overall, the combined fertilization treatments significantly optimized the soil environment (*p* < 0.05). Both soil pH and bulk density remained highest in the CK treatment and lowest in the Y2E2 treatment, with significant reductions of 11.10% and 37.20%, respectively, compared to CK (*p* < 0.05). Soil organic carbon, available phosphorus, and available potassium contents all reached their maximum levels in the Y2E2 treatment, showing significant increases of 92.94%, 99.28%, and 96.08%, respectively, compared to CK (*p* < 0.05), demonstrating its comprehensive effectiveness in improving soil fertility. Alkali-hydrolyzable nitrogen content ranged from 180.01 to 268.67 mg·kg^−1^, being lowest in the CK treatment and highest in the Y2E3 treatment (49.25% significant increase compared to CK, *p* < 0.05), with the Y2E2 treatment ranking second (257.37 mg·kg^−1^).

In conclusion, the combined fertilization treatments significantly optimized soil physicochemical properties by reducing soil pH and bulk density while enhancing available nutrient contents, with the Y2E2 treatment demonstrating optimal performance across all parameters, indicating its high efficacy for improving soil fertility and supporting grassland ecological restoration in alpine mining areas.

### 2.4. Effects of Combined Microbial Inoculant and Organic Fertilizer Application on Soil Enzyme Activities in Artificial Grassland

As shown in Figure 3a,b, soil β-glucosidase and dehydrogenase activities reached their highest levels under the Y2E2 treatment, measuring 127.66 U·g^−1^ and 21.87 U·g^−1^, respectively, representing significant increases of 77.15% and 66.61% compared to the control (CK) treatment (*p* < 0.05). Soil N-acetyl-glucosaminidase activity ranged from 52.70 to 94.09 U·g^−1^, with the highest activity observed in the Y2E3 treatment, followed by Y3E2, both showing significant increases of 78.52% and 78.51%, respectively, compared to CK (*p* < 0.05). Soil urease and acid phosphatase activities ranged from 1275.27 to 2098.15 U·g^−1^ and 33.35 to 91.83 μmol·d^−1^·g^−1^, respectively, with both enzymes exhibiting peak activity under the Y2E2 treatment, significantly higher than CK (*p* < 0.05). Soil protease (S-PT) activity was highest in the Y3E1 treatment (0.83 μmol·d^−1^·g^−1^), reflecting a 55.00% significant increase over CK (*p* < 0.05).

Figure 3c,d demonstrated that in 2024, the CK treatment still exhibited significantly lower soil enzyme activities than all combined fertilization treatments, confirming the persistence of the enhancement effects. Soil β-glucosidase and dehydrogenase activities ranged from 73.82 to 131.65 U·g^−1^ and 14.44 to 22.47 U·g^−1^, respectively, with both enzymes showing the highest activities under the Y2E2 treatment (78.33% and 55.69% increases over CK, respectively; *p* < 0.05). Soil N-acetyl-glucosaminidase activity was highest in the Y3E2 treatment (95.67 U·g^−1^), followed closely by Y2E3 (95.37 U·g^−1^), representing significant increases of 74.93% and 74.40%, respectively, compared to CK (*p* < 0.05). Soil urease and acid phosphatase activities peaked under the Y2E2 treatment at 2075.80 U·g^−1^ and 91.74 μmol·d^−1^·g^−1^, respectively, both significantly higher than CK (*p* < 0.05). Soil protease activity ranged from 0.59 to 0.84 μmol·d^−1^·g^−1^, with the highest activity in the Y3E2 treatment, followed by Y2E2, both showing significant increases of 42.13% and 41.57%, respectively, compared to CK (*p* < 0.05).

Thus, the Y2E2 fertilization treatment consistently enhanced key soil enzyme activities (β-glucosidase, dehydrogenase, urease, and acid phosphatase), optimizing soil functionality and reinforcing biochemical activity in the soil system.

### 2.5. Effects of Combined Microbial Inoculant and Organic Fertilizer Application on Soil Microbial Biomass Carbon, Nitrogen, and Phosphorus in Artificial Grassland

The 2023 data (Figure 4a–c) demonstrated that the control (CK) treatment showed the lowest contents of soil microbial biomass carbon, nitrogen, and phosphorus, while the combined fertilization treatments significantly increased all parameters. Specifically, microbial biomass carbon and microbial biomass nitrogen reached the highest levels in the Y2E2 treatment (391.28 mg·kg^−1^ and 54.56 mg·kg^−1^, respectively), representing significant increases of 114.84% and 147.35% compared to CK (*p* < 0.05). Microbial biomass phosphorus content ranged from 2.50 to 4.92 mg·kg^−1^, being highest in the Y2E3 treatment, followed by Y3E1, with the CK treatment showing significantly lower values than all other combined fertilization treatments (*p* < 0.05).

The 2024 data (Figure 4d–f) confirmed that the enhancement effects of combined fertilization on soil microbial biomass exhibited both persistence and stability. Soil microbial biomass carbon and phosphorus contents ranged from 185.83 to 404.45 mg·kg^−1^ and 2.56 to 4.98 mg·kg^−1^, respectively, with both parameters peaking in the Y2E2 treatment (117.64% and 94.78% increases versus CK, respectively; *p* < 0.05). Soil microbial biomass nitrogen content was highest in the Y2E3 treatment (54.02 mg·kg^−1^), followed closely by Y2E2 (53.84 mg·kg^−1^), while the CK treatment showed the lowest value (22.86 mg·kg^−1^), being significantly inferior to all other treatments.

These results demonstrated that the Y2E2 fertilization treatment consistently dominated microbial biomass enhancement across both experimental years, providing stable energy and nutrient supplies for soil microbial communities and thereby strengthening the functional stability of the soil ecosystem.

### 2.6. Correlation Analysis of Forage Yield Quality, Soil Physicochemical Properties, and Soil Enzyme Activities Under Different Fertilization Treatments

As shown in Figure 5a, aboveground biomass of artificial grassland exhibited highly significant correlations (*p* < 0.01) with soil physicochemical properties and microbial biomass carbon, nitrogen, and phosphorus. Belowground biomass showed significant correlation with soil bulk density (BD) (*p* < 0.05) and reached highly significant correlation levels (*p* < 0.01) with all other factors. Soil pH and bulk density collectively demonstrated significant inhibitory effects, both showing significant negative correlations (*p* < 0.05), which were stronger with soil available nutrients (alkali-hydrolyzable, available phosphorus, available potassium) and microbial biomass (carbon, nitrogen, and phosphorus). Notably, soil pH exhibited an extremely significant negative correlation with organic carbon (*p* < 0.001), confirming the pronounced inhibitory effect of alkaline environments on organic carbon accumulation.

Regarding nutrient cycling mechanisms, available phosphorus showed a highly significant positive correlation with microbial biomass phosphorus (*p* < 0.01), revealing the microbial-driven characteristics of phosphorus cycling. Other available nutrients (alkali-hydrolyzable, available potassium) not only displayed extremely significant positive correlations with microbial biomass carbon and nitrogen (*p* < 0.001) but also formed synergistic effects with soil organic carbon. This indicates that nitrogen and potassium availability enhances microbial biomass accumulation, thereby driving the turnover efficiency of soil carbon and nitrogen.

Figure 5b revealed that among forage nutritional quality indicators, both crude protein and soluble sugar contents exhibited highly significant correlations (*p* < 0.01) with soil physicochemical properties and microbial biomass carbon, nitrogen, and phosphorus. Ether extract content showed significant correlation with soil bulk density (*p* < 0.05) and reached highly significant correlation levels (*p* < 0.01) with all other factors. Furthermore, crude ash, acid detergent fiber, and neutral detergent fiber contents all demonstrated highly significant correlations (*p* < 0.01) with soil physicochemical properties and microbial biomass carbon, nitrogen, and phosphorus, confirming the comprehensive regulatory effects of soil physicochemical environment and microbial biomass on forage nutritional quality.

Figure 5c demonstrated that soil enzyme activities, including β-glucosidase, N-acetyl-glucosaminidase, urease, protease, and acid phosphatase, exhibited highly significant correlations (*p* < 0.01) with soil physicochemical properties and microbial biomass carbon, nitrogen, and phosphorus. In contrast, soil dehydrogenase showed significant correlation with soil bulk density (*p* < 0.05) and reached highly significant correlation levels (*p* < 0.01) with all other factors.

Consequently, this study confirmed that soil physicochemical properties systematically and significantly influence artificial grassland biomass, forage nutritional quality, and soil enzyme activities, providing theoretical support for optimizing fertilization strategies. Through targeted regulation of soil environments, it is possible to simultaneously enhance forage yield and quality while improving soil ecological service functions.

### 2.7. Path Analysis of Soil Physicochemical Properties, Microbial Biomass, Soil Enzyme Activities and Forage Yield Quality

This study employed structural equation modeling to elucidate the integrated regulatory mechanisms of different fertilization treatments on the soil-forage system (Figure 6). The model fit indices demonstrated excellent goodness-of-fit (Fisher’s C = 10.23, *p* = 0.37, *df* = 4, AIC = 56.23, BIC = 104.40). The results revealed that fertilization treatments exerted extremely significant positive direct effects on forage biomass parameters (*β* = 0.27, *p* < 0.001, R^2^ = 0.93) and similarly strong positive direct effects on forage nutritional quality indicators (*β* = 0.73, *p* < 0.001, R^2^ = 0.53). Fertilization treatments significantly drove alterations in soil physicochemical properties (*β* = 0.45, *p* < 0.001), which subsequently generated highly significant positive direct effects on microbial biomass (*β* = 0.73, *p* < 0.001) while exhibiting negative indirect influences on soil enzyme activities through microbial biomass. Soil physicochemical properties demonstrated strong positive direct effects on soil enzyme activities (*β* = 0.83, *p* < 0.001). Moreover, soil physicochemical properties showed extremely significant positive direct effects on both grassland biomass (*β* = 0.90, *p* < 0.001). Microbial biomass, as a crucial mediating variable, exerted significant positive direct effects on both forage biomass (*β* = 0.23, *p* < 0.05) and nutritional quality (*β* = 0.23, *p* < 0.05).

These results systematically elucidated the cascading regulatory network of fertilization–soil physicochemical properties–microbial community–forage production, demonstrating that fertilization management not only directly enhanced forage yield and nutritional quality but also indirectly influenced system functionality by modulating soil physicochemical properties and microbial biomass, thereby providing important theoretical foundations for sustainable grassland ecosystem management.

### 2.8. Comprehensive Evaluation of Soil Physicochemical Properties, Microbial Biomass, Soil Enzyme Activities, and Forage Yield Quality Characteristics

Principal component analysis effectively reduced the dimensionality of 23 measured parameters—encompassing yield quality traits, soil physicochemical properties, microbial biomass, and soil enzyme activities—into two orthogonal principal components (Figure 7a), with a cumulative variance explanation rate of 88.90% (Figure 7a). The first principal component (PC1) exhibited an eigenvalue of 19.71, accounting for 85.70% of total data variation, while PC2 contributed an additional 3.20% variance, confirming systematic interrelationships among indicators.

The composite score analysis revealed that the Y2E2 treatment achieved optimal performance along PC1 (score = 1.24), followed by Y2E3 (0.94). In contrast, Y3E2 and Y1E2 demonstrated superior responses along PC2, with respective scores of 1.62 and 1.32. The weighted comprehensive evaluation ranked the restoration efficacy of fertilization treatments in alpine mining area artificial grasslands in descending order: Y2E2 > Y2E3 > Y3E1 > Y3E2 > Y3E3 > Y2E1 > Y1E2 > Y1E3 > Y1E1 > CK, with Y2E2 showing the highest rehabilitation performance and CK demonstrating the lowest efficacy.

## 3. Discussion

### 3.1. Effects of Combined EM Microbial Inoculant and Organic Fertilizer Application on Productivity and Nutritional Quality of Artificial Grasslands

Forage yield, as a key indicator of grassland ecosystem structure and function, effectively reflects vegetation responses to fertilization management and restoration efficacy in degraded grasslands. The results of this study align with findings from Ren et al. [17] and Gao et al. [18], confirming that combined microbial inoculant and organic fertilizer application significantly enhances productivity in alpine mining area artificial grasslands. The Y2E2 treatment demonstrated optimal growth-promoting effects, increasing aboveground biomass by 75.97% and 68.88% compared to the control (CK) in 2023 and 2024, respectively, with corresponding belowground biomass increases of 36.05% and 35.53%. Notably, biomass showed consistent interannual growth trends. This synergistic effect primarily stems from functional microorganisms (e.g., nitrogen-fixing and phosphate-solubilizing bacteria) in the EM inoculant, working in concert with organic fertilizer. The inoculant activates soil nutrients through phosphorus/potassium solubilization and nitrogen fixation, while organic fertilizer improves soil structure (e.g., porosity enhancement, water-holding capacity) to optimize root growth environments and nutrient acquisition [19]. Their combined action enhances photosynthetic efficiency in shoots while strengthening root growth and nutrient capture capabilities, collectively driving biomass accumulation [20]. Crucially, this study identified a clear threshold effect: the Y3E3 treatment showed reduced vegetation growth metrics compared to Y2E2, suggesting potential saturation or threshold effects for EM inoculant dosage or microbial density, consistent with Song et al. [21].

The combined application of EM microbial inoculant and organic fertilizer significantly improved forage nutritional quality, consistent with findings from Da et al. [22] and Montañez et al. [23]. The Y2E2 treatment demonstrated consistently superior effects across both experimental years, with crude protein and ether extract contents increasing by 46.18%~46.52% and 62.42%~63.25%, respectively, compared to the control (CK), while soluble sugar content rose by 19.49~20.56%. The observed improvements in crude protein, a key nutritional indicator, primarily resulted from nitrogen-fixing bacteria in the EM inoculant significantly enhancing soil available nitrogen content, thereby providing sufficient nitrogen sources for plant protein synthesis. This aligns with findings by Akladious et al. [24] in maize cultivation, where microbial inoculation promoted nitrogen assimilation processes and subsequently increased crude protein content. Concurrently, enhanced activity of phosphate-solubilizing bacteria improved phosphorus availability. As a critical component of ATP and phospholipids, phosphorus directly participates in lipid synthesis pathways, explaining the EE content elevation [25]. Furthermore, the organic carbon supplied by organic fertilizers fueled microbial activity, increasing nutrient availability and promoting the accumulation of photosynthetic products like soluble sugar, corroborating Liu et al.’s observations [26]. The reduced crude ash content in our study relates to organic fertilizers’ regulatory effects on mineral element uptake, mirroring Mbuthia et al.’s results with cattle manure applications [27]. This suggests that the inoculant–fertilizer combination improves nutrient absorption efficiency by optimizing soil aeration and other environmental conditions, ultimately enhancing forage digestibility [25].

### 3.2. Effects of EM Microbial Inoculant and Organic Fertilizer Co-Application on Soil Physicochemical Properties, Enzyme Activities, and Microbial Biomass

The combined application of microbial inoculants and organic fertilizers has been established as an effective strategy for improving degraded soil quality through three primary mechanisms: enhancement of soil nutrient retention capacity, elevation of soil enzyme activities, and increases in microbial biomass carbon, nitrogen, and phosphorus [26]. The results of this study demonstrated significant agreement with previous studies conducted by Mbuthia [27] and Lin et al. [28], providing robust evidence for the synergistic soil amelioration effects achieved through the combination of microbial inoculants and organic fertilizers. Among all treatments, Y2E2 exhibited the most pronounced soil improvement effects. With respect to soil pH regulation, the Y2E2 treatment resulted in pH reductions of 8.31% and 11.10% compared to the control (CK) in 2023 and 2024, respectively. These observations are consistent with the mechanistic studies of Wang et al. [29] on organic acid neutralization, confirming that organic acids generated during fertilizer decomposition effectively neutralize alkaline ions in the soil system. Regarding soil structural improvements, the Y2E2 treatment significantly reduced soil bulk density by 37.20% in 2024, which aligns with the findings of Shi et al. [30] concerning organic matter-induced aggregate formation. The cementing action of organic matter substantially increased the proportion of water-stable aggregates exceeding 0.25 mm in diameter, thereby improving both soil pore structure and water-holding capacity. These improvements in soil physical properties create more favorable conditions for root growth and microbial activity, which are essential for sustainable grassland ecosystem functioning.

From the perspective of soil nutrient supply, the Y2E2 treatment significantly improved soil nutrient availability, with soil organic carbon, alkaline hydrolyzable nitrogen, and available potassium increasing by 72.72%, 50.98%, and 96.88%, respectively, in 2023. These values further rose to 92.94%, 49.25%, and 96.08% in 2024, demonstrating sustained nutrient enrichment. These results markedly exceeded the effects of sole organic fertilizer application reported by Zhi et al. [31], underscoring the synergistic activation effect of EM microbial inoculants. Specifically, EM inoculants facilitated the conversion of recalcitrant organic phosphorus, mineral-bound potassium, and immobilized nitrogen into plant-available forms through phosphorus solubilization, potassium release, and biological nitrogen fixation. Concurrently, organic fertilizers functioned as a “nutrient reservoir,” gradually mineralizing organic carbon and releasing essential macro- and micronutrients to support sustained plant growth [32]. The native soils in alpine mining areas are characterized by alkaline pH, high bulk density, and nutrient deficiency. The combined fertilization approach significantly improved these soil properties, establishing favorable conditions for root growth and nutrient uptake in artificial grasslands. These improvements represent a critical prerequisite for ecological restoration of degraded grasslands in these environments.

Soil enzymes serve as key drivers of organic matter decomposition and biogeochemical nutrient cycling, with their activities directly reflecting soil biochemical functional capacity [33]. Microbial inoculants primarily enhance enzyme activities through two established mechanisms: (1) regulation of microbial community structure (abundance and activity), and (2) improvement of soil nutrient microenvironments, as validated across multiple ecosystems [34]. This study further demonstrates that EM microbial inoculant–organic fertilizer co-application significantly enhances key soil enzyme activities in alpine mining area artificial grasslands, with the Y2E2 treatment showing particularly stable and sustained effects, consistent with findings by Ren et al. [35] in alpine grasslands. Enzyme activity dynamics revealed substantial improvements in 2023 under the Y2E2 treatment compared to the control (CK): β-glucosidase (77.15%), dehydrogenase (66.61%), urease (64.28%), and acid phosphatase (175.35%). These enhancements remained stable or increased further in 2024 (78.33%, 55.69%, 55.87%, and 183.67%, respectively), surpassing the results reported by Maciejewska et al. [36] and confirming the unique synergistic effects of EM inoculants. Notably, the acid phosphatase activity increase substantially exceeded the effects of organic and inorganic fertilization amendments reported by Chen et al. [37], highlighting the superior phosphorus activation capacity of this co-application strategy. Mechanistic analysis revealed dual pathways for EM-mediated enzyme activity enhancement: (1) Direct pathway: Inoculant-contained Bacillus spp. and Trichoderma spp. directly secrete key enzymes including β-glucosidase and acid phosphatase, aligning with Chen et al.’s findings on microbial enzyme secretion characteristics [38]; (2) Indirect pathway: Organic fertilizer-derived carbon sources stimulate native microbial growth and enzyme synthesis capacity through the priming effect, corroborating the carbon priming theory proposed by Zhou et al. [39].

Microbial biomass, a fundamental biological indicator of soil fertility, serves both as a direct measure of microbial abundance and activity and as an active nutrient reservoir in soil ecosystems. Characterized by significantly faster turnover rates than soil organic matter, microbial biomass plays a crucial regulatory role in plant nutrient supply [40]. These results demonstrate that the co-application of EM microbial inoculant and organic fertilizer significantly enhanced soil microbial biomass in alpine mining area artificial grasslands, with the Y2E2 treatment exhibiting optimal effects. These findings align with the microbial inoculant synergy mechanisms reported by Su et al. [41]. Quantitative analysis revealed that Y2E2 treatment induced the most pronounced stimulatory effects, increasing soil microbial biomass carbon and nitrogen by 114.84% and 147.35%, respectively, in 2023 compared to the control (CK). Furthermore, sustained enhancements were observed in 2024, with microbial biomass carbon and microbial biomass phosphorus increasing by 117.64% and 94.78%, respectively. These improvements substantially surpassed the effects of conventional fertilization practices documented by Yin et al. [42], confirming the unique synergistic advantages of microbial–organic matter interactions. The underlying mechanisms primarily involved the EM inoculants introducing beneficial microorganisms (e.g., nitrogen-fixing bacteria, phosphate-solubilizing bacteria), directly augmenting microbial populations. Organic fertilizers supplied abundant carbon substrates (e.g., carbohydrates, proteins) and mineral nutrients (e.g., N, P), creating optimal conditions for microbial growth and proliferation [41]. Notably, increased microbial biomass established a positive feedback loop with improved soil physicochemical properties and enzyme activities: enhanced microbial biomass promoted enzyme synthesis and accelerated nutrient cycling, while optimized soil conditions further stimulated microbial growth [43]. This self-reinforcing cycle significantly strengthened the functional stability of soil ecosystems, providing sustainable edaphic support for long-term grassland productivity and ecological services (e.g., forage production, soil conservation).

### 3.3. Correlation Analysis and Comprehensive Evaluation of Forage Productivity, Nutritional Quality, and Soil Biochemical Properties

Correlation analysis identified soil pH and bulk density as key inhibitory factors for soil fertility, showing significant negative correlations with available nutrients (alkali-hydrolyzable nitrogen, available phosphorus, available potassium) and increases in microbial biomass carbon, nitrogen, and phosphorus. Particularly, pH demonstrated extremely significant suppression of soil organic carbon, consistent with findings by Jing et al. [44] and Ma et al. [45], confirming alkaline conditions as critical constraints for carbon accumulation and nutrient availability in alpine mining ecosystems. Microbial biomass emerged as a pivotal mediator between soil properties and plant and enzyme activities, with its strong positive correlation with available phosphorus highlighting microbial-driven phosphorus cycling. The synergistic relationships among nitrogen and potassium availability, microbial biomass, and organic carbon indicated microbial-mediated nutrient mobilization that sustains carbon–nitrogen turnover and forage production [40].

Structural equation modeling elucidated fertilization’s dual effects: (1) direct stimulation of forage biomass and nutritional quality, and (2) indirect regulation through soil property modifications that enhanced microbial biomass and enzyme activities [41]. The significant mediation effect of microbial biomass validated the soil improvement–microbial activation–nutrient supply cascade as the principal pathway for grassland productivity enhancement [46]. Principal component analysis (cumulative variance explained: 88.90%) and composite scoring demonstrated the Y2E2 treatment’s superior performance in optimizing soil properties, microbial parameters, and forage quality indices, significantly outperforming other treatments. These findings systematically confirm that targeted regulation of soil pH, structural improvement, and microbial enzymatic activation constitute the core strategy for synergistic enhancement of forage production and ecological functions. The study establishes a mechanistic framework for soil–microbe–plant interactions in degraded alpine ecosystems, with Y2E2 formulation representing the optimal fertilization strategy for sustainable grassland restoration.

### 3.4. Cost-Effectiveness and Practical Feasibility

The Y2E2 treatment, combining moderate rates of EM microbial inoculant and organic fertilizer, demonstrated not only optimal agronomic performance but also favorable cost-effectiveness and field applicability. Compared to high-dose applications (e.g., Y3E3), Y2E2 achieved maximum gains in forage yield, nutritional quality, and soil health at a lower input cost, avoiding the diminishing returns associated with excessive inoculant or fertilizer use. The EM inoculant is relatively low-cost, scalable for mass production, and compatible with existing fertilizer application methods, enabling easy integration into current grassland management practices [47]. Furthermore, organic fertilizers used in this study can be sourced locally from livestock operations in alpine regions, reducing transportation and input costs while promoting circular agriculture. The significant improvement in soil properties and sustained productivity over two years also suggests reduced long-term input dependency, lowering maintenance costs for grassland restoration. Given its technical simplicity, environmental adaptability, and strong ecological benefits, the Y2E2 strategy presents a feasible and economically viable approach for large-scale rehabilitation of degraded alpine mining ecosystems.

## 4. Materials and Methods

### 4.1. General Situation of the Study Area

The research area is located in the alpine ecotone of the Northeastern Qinghai-Tibet Plateau (38°9′34″ N, 99°9′40″ E), within the Muli mining district of Tianjun County, Haixi Mongolian and Tibetan Autonomous Prefecture, Qinghai Province. The study region spans approximately 400 km^2^ with a mean elevation of 4000 m, characterized by a typical plateau mountain climate featuring ecological fragility, indistinct seasonal boundaries, low mean annual temperature (−5 °C), and significant diurnal temperature variations [2], detailed information is provided in Figure 8.

The study area exhibits pronounced seasonal heterogeneity in hydrothermal conditions, characterized by a concentrated snowfall period from November to May, with a mean annual precipitation of 277 mm. The predominant soil parent materials consist of alpine meadow soil and swamp meadow soil. The dominant vegetation includes *Carex tibetikobresia* S. R. Zhang and *Carex moorcroftii* Falc. ex Boott of the Cyperaceae family, as well as *Bistorta vivipara* (L.) Gray of the Polygonaceae family [2].

Surface characteristics reveal widespread coverage of coal gangue and rock debris deposits on degraded meadow topsoils. Key soil physicochemical parameters include total nitrogen (TN) 1.05 g·kg^−1^, total phosphorus (TP) 0.84 g·kg^−1^, soil organic carbon (SOC) 60.34 g·kg^−1^, electrical conductivity (EC) 19,090 μS·cm^−1^, pH 8.50, and soil water content (SWC) 15.0% [2].

### 4.2. Experimental Materials

The experiment utilized a composite Effective Microorganisms (EM) microbial inoculant as the core microbial preparation. The functional microbial consortium comprised dominant Gram-positive *Bacillus* spp. strains (*Bacillus subtilis*, *Bacillus megaterium*, and *Bacillus mucilaginosus*), photosynthetic bacteria (Photosynthetic Bacteria), *Saccharomyces* spp., and *Lactobacillus* spp., with a viable bacterial count ≥ 1 × 10^8^ CFU·g^−1^. The inoculant was provided by Qiming Bioengineering Co., Ltd. (Hubei, China) and was produced following standardized industrial manufacturing processes. All technical specifications complied with the quality standards of Agricultural Microbial Inoculants (GB 20287–2006) [48].

The bio-organic fertilizer used in this experiment was a mineral-derived humic acid product with organic matter content ≥ 40% and total nutrient content (N+P_2_O_5_+K_2_O) ≥ 5.0%. Microbial safety parameters met the following standards: fecal coliforms ≤ 100 CFU·g^−1^ and ascarid egg inactivation rate ≥ 95%. Heavy metal concentrations complied with the Chinese Agricultural Industry Standard “Organic Fertilizers” (NY/T 525–2021) [49]: total arsenic (As) ≤ 15 mg·kg^−1^, total mercury (Hg) ≤ 2 mg·kg^−1^, total lead (Pb) ≤ 50 mg·kg^−1^, total cadmium (Cd) ≤ 3 mg·kg^−1^, and total chromium (Cr) ≤ 150 mg·kg^−1^. The fertilizer was industrially produced by Henan Liso Crop Protection Co., Ltd. (Zhengzhou, China).

Mineral-derived humic acid organic fertilizer is a composite organic fertilizer produced through alkaline, acidic, or organic solvent extraction of humic acids from raw materials including peat, lignite, weathered coal, and other low-rank coals combined with animal/plant residues, followed by incorporation with NPK and micronutrients. Its core functionalities encompass soil structure improvement, microbial activity enhancement, and nutrient availability promotion.

The experimental grass species consisted of *Festuca sinensis* Keng. cv. ‘Qinghai’, *Poa Pratensis* L. cv. ‘Qinghai’, *Poa crymophila* cv. ‘Qinghai’, and *Elymus breviaristatus* keng cv. ‘Tongde’. All seeds were provided by the Grassland Research Institute of the Academy of Animal Science and Veterinary Medicine, Qinghai University, with quality parameters meeting the Chinese National Standard “Quality Grading of Forage Seeds” (GB 6141–2008) [50]: purity ≥ 92% and germination rate ≥ 85%.

### 4.3. Experimental Design

Artificial Grassland Establishment: On 5 May 2022, site preparation was conducted using an SY335BH-S hydraulic excavator (Sany Heavy Industry, Beijing, China) to remove rock fragments and obstacles larger than 5 cm in diameter and establish a 20 cm cultivation layer. Subsequently, a 1BQD-3.4 disc harrow (Sany Heavy Industry, Beijing, China) was employed to thoroughly incorporate mine-derived humic acid organic fertilizer with spoil materials at a controlled tillage depth of 10 cm. The forage seeds were uniformly mixed at equal proportions and sown mechanically at a density of 18 g·m^−2^. Post-sowing operations included mechanical soil covering (3–5 cm thickness), hydraulic compaction, and surface application of biodegradable non-woven fabric (20 ± 2 g·m^−2^) with a water permeability rate ≥ 85%. The fabric was removed for environmentally safe disposal in late July when seedling coverage reached 80%.

Fertilization Treatments: Based on the previous research results of the team, soil fertility measurement data, and the nutrient requirement laws of artificial grasslands, an experiment on the combined application of organic fertilizer and EM microbial inoculant (EM) in different proportions was constructed. Three levels of organic fertilizer (10.0, 20.0, 40.0 t·hm^−2^) and three gradients of EM inoculant (0.45, 0.60, 0.75 t·hm^−2^) were set. A randomized block experimental design was adopted, with a total of 10 treatment combinations (Table 3). Among them, the control treatment (CK) only applied organic fertilizer (10.0 t·hm^−2^) without adding EM microbial inoculant. The area of each experimental plot was 20 m^2^ (4 m × 5 m), with a 2 m buffer isolation zone set. Each treatment was repeated 3 times, and a total of 30 plots were set. Fertilization operations were carried out in mid-June 2023 and 2024.

### 4.4. Plant and Soil Sampling

Forage Yield and Nutritional Quality Determination: Continuous field monitoring was conducted in September of both 2023 and 2024. Five 50 cm × 50 cm quadrats were randomly established in each experimental plot for sampling. Aboveground vegetation was harvested at ground level, with fresh weights recorded before subjecting samples to 105 °C de-enzyming for 30 min, followed by constant drying at 65 °C until reaching constant weight (48 ± 2 h) to determine aboveground biomass dry weight. Post-harvest, root samples were collected from 0~10 cm soil depth using a 10 cm diameter auger in a five-point sampling pattern. Composite root samples were washed through a 1.5 mm sieve and oven-dried at 65 °C to constant weight for belowground biomass quantification. Dried plant materials were pulverized using an ultra-fine grinder, sieved through 1 mm mesh, and stored in desiccators for subsequent analyses [51].

Nutritional composition was determined using standardized analytical methods: Crude protein (CP) content was determined by the Dumas method using high-temperature combustion (1150 °C) with a Vario EL Cube elemental analyzer (Elementar Analysensysteme GmbH, Langenselbold, Germany). Nitrogen content was calculated from nitrogen gas signal response calibrated with sulfanilamide (N = 11.59%), and CP was estimated as N × 6.25 [52].

Lipid content, expressed as ether extract (EE), was determined by exhaustive solvent extraction using a Soxtec 8000 automated system (FOSS Analytical, Hillerød, Denmark). Briefly, 1.5 g of ground sample was extracted with petroleum ether (boiling range 40–60 °C) over 30 min of heating, 60 min of reflux, and 30 min of recovery, followed by solvent evaporation. Extracted residues were dried at 105 °C for 1 h and weighed to determine lipid content [52].

Soluble sugars (SS), representing water-soluble carbohydrates, were quantified using the anthrone–sulfuric acid method. A 0.1 g subsample was extracted with 80% (*v*/*v*) hot ethanol (80 °C, 3 × 15 min), and the combined extracts were reacted with anthrone reagent (0.2%, *w*/*v* in concentrated H_2_SO_4_). Absorbance was measured at 620 nm using a UV-2600 dual-beam spectrophotometer (Shimadzu, Kyoto, Japan), with glucose used as the calibration standard (0–100 μg·mL^−1^) [52].

Crude ash content was determined via complete oxidation of organic matter. Approximately 2 g of sample was ignited in pre-weighed porcelain crucibles in a Thermolyne F6010 programmable muffle furnace (Thermo Fisher Scientific, Waltham, MA, USA) at 550 ± 5 °C for 6 h, cooled in a desiccator, and re-weighed. Ash content was calculated as the mass loss after combustion relative to the initial dry weight [52].

Neutral detergent fiber (NDF) and acid detergent fiber (ADF) were determined according to the sequential Van Soest method. NDF was analyzed using an ANKOM A200 fiber analyzer (ANKOM Technology, Macedon, NY, USA) with heat-stable α-amylase (Sigma-Aldrich, A3306) added to neutral detergent solution (CTAB, Na_2_EDTA, Na_2_CO_3_, sodium lauryl sulfate) to prevent protein precipitation, and samples were incubated at 100 °C for 1 h. Residues were filtered under vacuum using pre-ashed and pre-weighed glass fiber filter bags (ANKOM F57), rinsed with hot water and acetone, and dried at 105 °C. ADF was determined via further digestion of NDF residues in acid detergent solution (H_2_SO_4_, cetyltrimethylammonium bromide) at 100 °C for 1 h, followed by filtration, washing, and drying. Acid detergent lignin (ADL) was estimated as the residue after 72 h of ADF digestion with 72% H_2_SO_4_ at 20 °C [52].

Soil Sample Collection: Soil samples were collected synchronously using a five-point composite sampling method. A 5 cm diameter soil auger was employed to extract samples from the 0~10 cm soil layer, with individual plot samples homogenized into one composite sample. Fresh soil was sieved (≤2 mm) to remove roots and gravel, then divided into two subsamples: (1) fresh samples stored at 4 °C in darkness (≤24 h) for soil enzyme activity and microbial biomass carbon/nitrogen analysis; and (2) air-dried samples (stored in a cool, ventilated area) that were ground and passed through a 0.25 mm sieve for routine physicochemical analysis [53].

Soil bulk density (*BD*) was determined in the 0~10 cm soil layer using the core method with a stainless-steel cutting ring (volume: 100 cm^3^; diameter: 5.08 cm; height: 5.0 cm). The ring was carefully inserted vertically into the soil using a hammer and driving cap to minimize compaction, ensuring undisturbed core collection. Excess soil at both ends was trimmed with a sharp knife, and loose surface material was removed. Wet soil mass (*M_wet_*) was recorded, then oven-dried to constant weight (105 °C, 8 h) to determine dry mass (*M_dry_*). *BD* was calculated as follows [53]:(1)$$BD=\rho \times \frac{Mwet}{Mdry}$$

### 4.5. Measurements and Methods

Soil Physicochemical Properties Analysis: Soil pH was determined potentiometrically using a benchtop pH meter (Seven Compact S220, Mettler Toledo, Switzerland) in a 1:2.5 (*w*/*v*) soil-to-water suspension after equilibration for 30 min with continuous stirring. Soil organic carbon (SOC) was determined by potassium dichromate oxidation: 0.25 g of air-dried soil was digested with 0.167 mol·L^−1^ K_2_Cr_2_O_7_ and concentrated H_2_SO_4_, heated at 170–180 °C for 5 min, and titrated with 0.1 mol·L^−1^ FeSO_4_; a blank correction was applied. Alkali-hydrolyzable nitrogen (AN) was determined via the alkaline diffusion method with a diffusion dish and titration of released ammonia using 0.01 mol·L^−1^ H_2_SO_4_. Available phosphorus (AP) was extracted with 0.5 mol·L^−1^ NaHCO_3_ (Olsen method) at pH 8.5 and quantified using the molybdenum–antimony colorimetric method at 880 nm. Available potassium (AK) was extracted with 1 mol·L^−1^ ammonium acetate (NH_4_OAc) at pH 7.0 and measured using flame photometry (FP 640, Shanghai Precision & Scientific Instrument Co., Shanghai, China) [53].

Soil enzyme activities related to carbon, nitrogen, and phosphorus cycling: Soil β-glucosidase (S-βBG) activity was assayed using p-nitrophenyl-β-D-glucopyranoside (pNPG) as substrate; enzyme activity was determined spectrophotometrically at 400 nm after incubation at 37 °C for 1 h. Soil dehydrogenase (S-DHA) activity was measured based on the reduction of triphenyltetrazolium chloride (TTC) to triphenylformazan (TPF), which was extracted with methanol and quantified at 480 nm. Soil N-acetyl-β-glucosaminidase (S-NAG) activity was determined using p-nitrophenyl-N-acetyl-β-D-glucosaminide (pNAG) as substrate and measured at 400 nm. Soil urease (S-UE) activity was assessed using indophenol blue colorimetry after incubation with urea solution at 37 °C for 24 h, and absorbance was read at 630 nm. Soil protease (S-PT) activity was determined using L-leucine as substrate and quantified via the ninhydrin colorimetric method at 570 nm. Soil acid phosphatase (S-AP) activity was measured via hydrolysis of p-nitrophenyl phosphate (pNPP) disodium salt at pH 6.5, with p-nitrophenol release detected at 400 nm. All enzyme assays were conducted in triplicate with appropriate controls (substrate-free and killed-soil blanks) [54].

Soil microbial biomass: Microbial biomass carbon (MBC) was determined using the chloroform fumigation–K_2_SO_4_ extraction method. Briefly, one set of soil samples was fumigated with ethanol-free chloroform for 24 h at 25 °C, while the control set was not fumigated. Both sets were extracted with 0.5 mol·L^−1^ K_2_SO_4_, and organic carbon in the extracts was measured using a total organic carbon (TOC) analyzer (TOC-L, Shimadzu, Japan). MBC was calculated using a conversion factor (kC = 0.45). Microbial biomass nitrogen (MBN) was determined similarly via fumigation–extraction, with total nitrogen in the extract measured using the Kjeldahl method and calculated using a kN factor of 0.54. Microbial biomass phosphorus (MBP) was determined via chloroform fumigation–0.5 mol·L^−1^ NaHCO_3_ extraction (pH 8.5), with inorganic phosphorus in the extract measured colorimetrically at 700 nm; a kP factor of 0.40 was applied [54].

### 4.6. Data Analysis and Visualization

The geographical location map of the experimental area was generated using ArcGIS (ESRI 10.2). Data organization was performed in Microsoft Excel 2016. Treatment effects on forage yield and quality, soil physicochemical properties, enzyme activities, and microbial biomass were evaluated using one-way analysis of variance (ANOVA, SPSS 27.0). Where ANOVA indicated significant differences (*p* < 0.05), Duncan’s multiple range test was applied to identify specific treatment differences (significance threshold α = 0.05). Factor analysis was conducted to extract principal components with eigenvalues > 1, with linear combination equations constructed based on eigenvectors. Principal component weights were determined when cumulative contribution rates exceeded 85%, followed by comprehensive score calculations.

A fertilization–soil–plant structural equation model (SEM) was developed using the lavaan package (v0.6-16) in R. Data were standardized via Z-score transformation to eliminate dimensional effects, with path coefficients estimated using the maximum likelihood (ML) method. Model covariance matrix fitness was verified using Fisher’s C test, while optimal model architecture was selected based on Akaike (AIC) and Bayesian (BIC) information criteria. The final model quantified effect strengths through standardized path coefficients (β, significance threshold *p* < 0.05), with SEM path diagrams generated using the semPlot (version 1.1.6) package. Pearson correlation matrices were visualized as heatmaps via the corrplot package. Remaining figures were produced in Origin 2022.

## 5. Conclusions and Prospects

### 5.1. Conclusions

This two-year field study (2023–2024) demonstrates that the co-application of EM microbial inoculant (0.60 t·hm^−2^) and organic fertilizer (20 t·hm^−2^; Y2E2 treatment) represents the optimal fertilization strategy for artificial grassland restoration in alpine mining areas. The Y2E2 treatment significantly increased aboveground biomass by 75.97% (2023) and 68.88% (2024) and belowground biomass by 36.05% (2023) and 35.53% (2024) while substantially improving forage quality through increased crude ether extract (62.42%~63.25%) and soluble sugar content (19.49%~20.56%), demonstrating enhanced nutritional characteristics with reduced fiber content. Soil amelioration results showed that Y2E2 treatment significantly reduced soil pH (11.10%) and bulk density (37.20%) while increasing organic carbon (92.94%), alkali-hydrolyzable nitrogen (49.25%), and available potassium (96.08%) by 2024 compared to the control (CK). Substantial improvements in key enzyme activities and microbial biomass confirmed the treatment’s capacity to stimulate microbial-mediated nutrient cycling. Structural equation modeling revealed strong direct positive effects of co-application fertilization on forage yield (*β* = 0.27, *p* < 0.001, R^2^ = 0.93) and quality (*β* = 0.73, *p* < 0.001, R^2^ = 0.53). These findings collectively demonstrate that EM–organic fertilizer co-application synergistically enhances soil–plant system functionality in alpine mining ecosystems, achieving simultaneous ecological restoration and forage productivity/quality improvement. This approach provides an efficient and sustainable technological pathway for artificial grassland rehabilitation in degraded alpine regions.

### 5.2. Prospects

While this study has established the rehabilitation efficacy and optimal application strategy for EM microbial inoculant–organic fertilizer co-application in alpine mining grasslands, several critical research directions warrant further investigation to advance both theoretical understanding and practical applications. A comprehensive long-term monitoring program should be implemented to evaluate treatment effects on ecosystem stability indicators, including soil carbon sequestration capacity and microbial community succession patterns, as well as system resilience to environmental perturbations such as extreme climate events and grazing pressure. A comprehensive economic feasibility analysis is particularly needed, including: (1) production costs of EM inoculants at industrial scale, (2) full cost accounting of organic fertilizer sourcing, transportation, and application, (3) cost–benefit comparisons for different application ratios of economic evaluations relative to conventional remediation approaches—these economic parameters will critically determine the large-scale implementation potential in remote alpine mining regions. These integrated investigations will significantly contribute to developing a robust theoretical framework for alpine ecosystem restoration while enhancing the precision, adaptability, and predictive capacity of rehabilitation technologies under varying environmental conditions.

## Figures and Tables

**Figure 1 plants-14-03156-f001:**
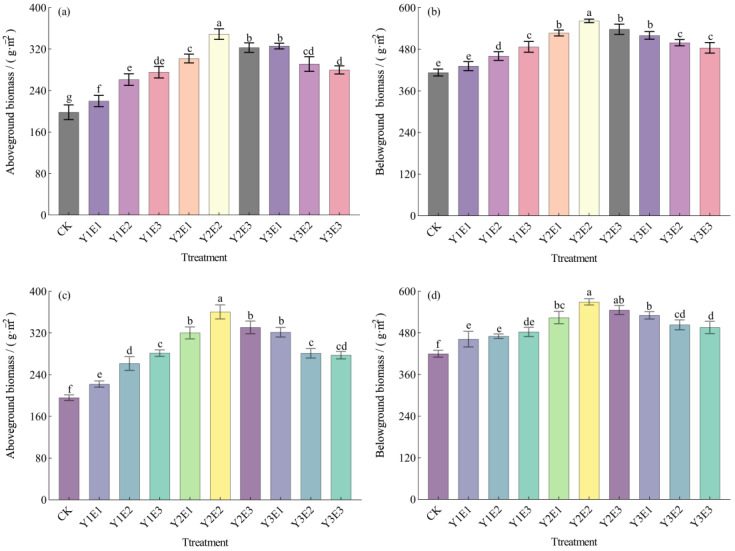
Effects of different fertilization treatments on aboveground and belowground biomass of artificial grassland vegetation. Panel (**a**) shows aboveground biomass in 2023; (**b**) belowground biomass in 2023; (**c**) aboveground biomass in 2024; (**d**) belowground biomass in 2024. Lowercase letters (e.g., “a”, “b”) indicate statistically significant differences among treatments at *p* < 0.05 according to Duncan’s multiple range test. Data are presented as mean ± standard deviation.

**Figure 2 plants-14-03156-f002:**
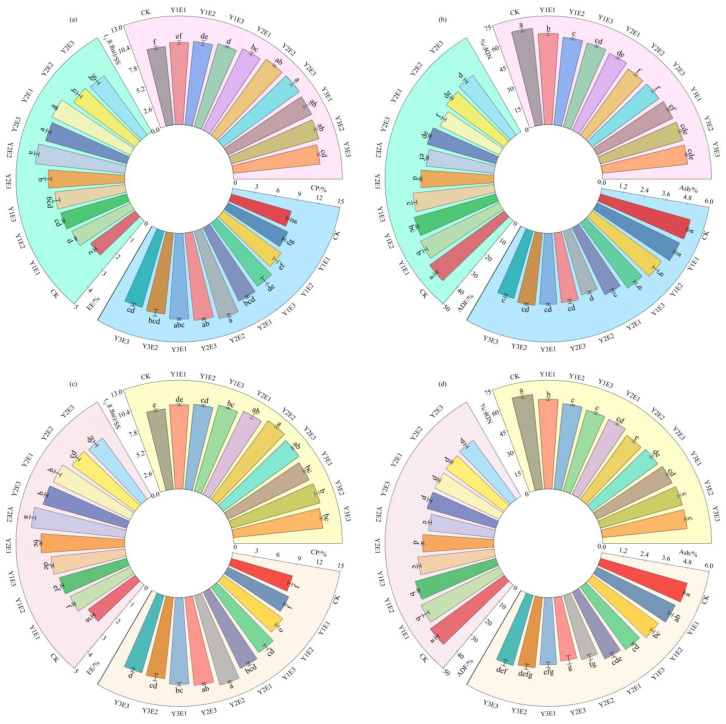
Effects of different fertilization treatments on forage nutritional quality in artificial grasslands. Panels (**a**,**b**) present nutritional quality analyses for 2023, while panels (**c**,**d**) show the 2024 results. CP: crude protein; EE: ether extract; SS: soluble sugars; Ash: crude ash; ADF: acid detergent fiber; NDF: neutral detergent fiber. Lowercase letters (e.g., “a”, “b”) denote significant differences among treatments at *p* < 0.05.

**Figure 3 plants-14-03156-f003:**
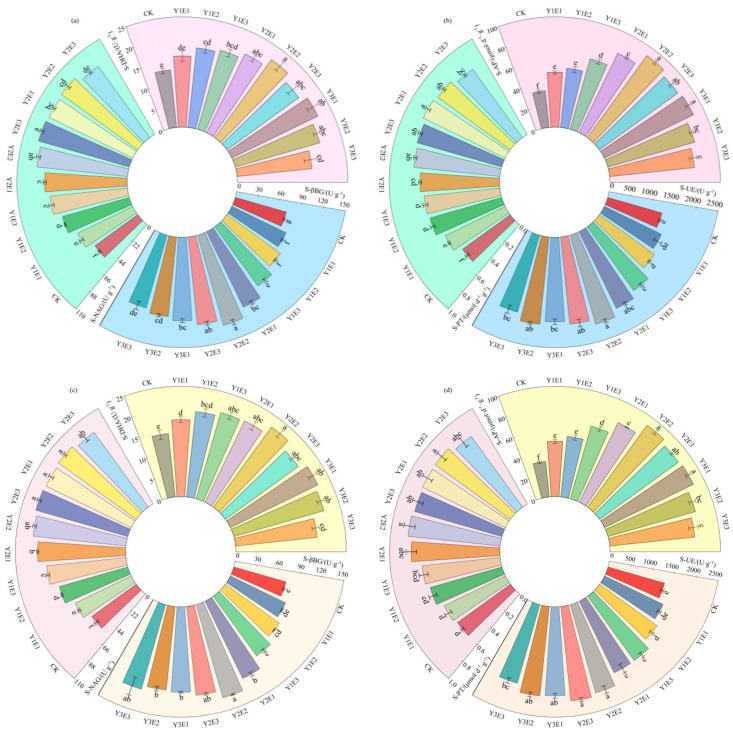
Effects of different fertilization treatments on soil enzyme activities in artificial grasslands. Panels (**a**,**b**) show the soil enzyme activity analyses for 2023, while panels (**c**,**d**) present the 2024 data. S-βBG: soil β-glucosidase; S-DHA: soil dehydrogenase; S-NAG: soil N-acetyl-β-D-glucosaminidase; S-UE: soil urease; S-PT: soil protease; S-AP: soil acid phosphatase. Lowercase letters (e.g., “a”, “b”) indicate statistically significant differences among treatments at *p* < 0.05 according to Duncan’s multiple range test.

**Figure 4 plants-14-03156-f004:**
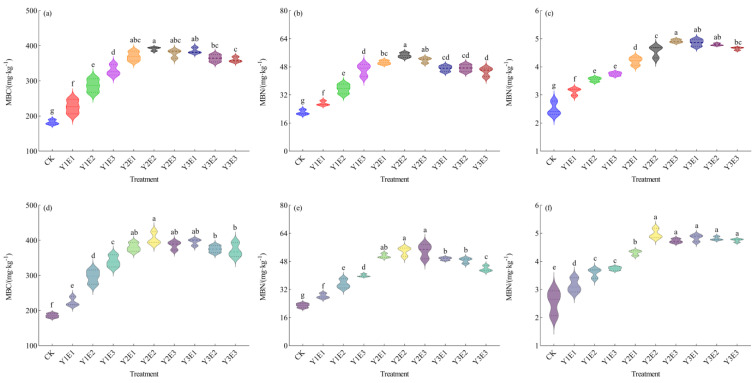
Effects of different fertilization treatments on soil microbial biomass carbon (MBC), nitrogen (MBN), and phosphorus (MBP) in artificial grasslands. Panels (**a**–**c**) present the 2023 analyses, while panels (**d**–**f**) display the 2024 results. MBC: soil microbial biomass carbon; MBN: soil microbial biomass nitrogen; MBP: soil microbial biomass phosphorus. Lowercase letters (e.g., “a”, “b”) denote statistically significant differences among treatments at *p* < 0.05 according to Duncan’s multiple range test.

**Figure 5 plants-14-03156-f005:**
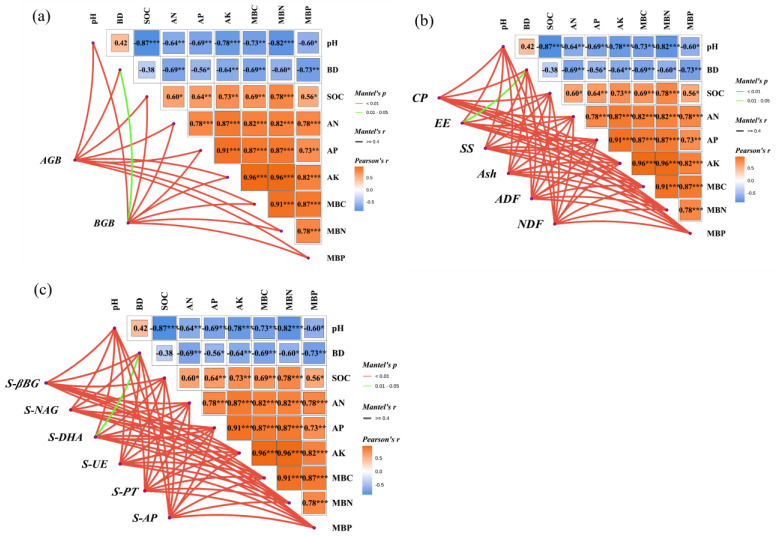
Correlation analyses of forage yield quality, soil physicochemical properties, and soil enzyme activities under different fertilization treatments. (**a**) Correlations between aboveground biomass and belowground biomass of artificial grassland with soil physicochemical properties and microbial biomass. (**b**) Correlations between forage nutritional quality indicators and soil physicochemical properties with microbial biomass. (**c**) Correlations between soil enzyme activities and soil physicochemical properties with microbial biomass. AGB: aboveground biomass; BGB: belowground biomass; pH: soil pH; BD: soil bulk density; SOC: soil organic carbon content; AN: alkali-hydrolyzable nitrogen content; AP: available phosphorus content; AK: available potassium content; MBC: soil microbial biomass carbon; MBN: soil microbial biomass nitrogen; MBP: soil microbial biomass phosphorus; CP: crude protein; EE: ether extract; SS: soluble sugars; Ash: crude ash; ADF: acid detergent fiber; NDF: neutral detergent fiber; S-βBG: soil β-glucosidase; S-DHA: soil dehydrogenase; S-NAG: soil N-acetyl-β-D-glucosaminidase; S-UE: soil urease; S-PT: soil protease; S-AP: soil acid phosphatase. Statistical significance levels: * *p* < 0.05, ** *p* < 0.01, *** *p* < 0.001.

**Figure 6 plants-14-03156-f006:**
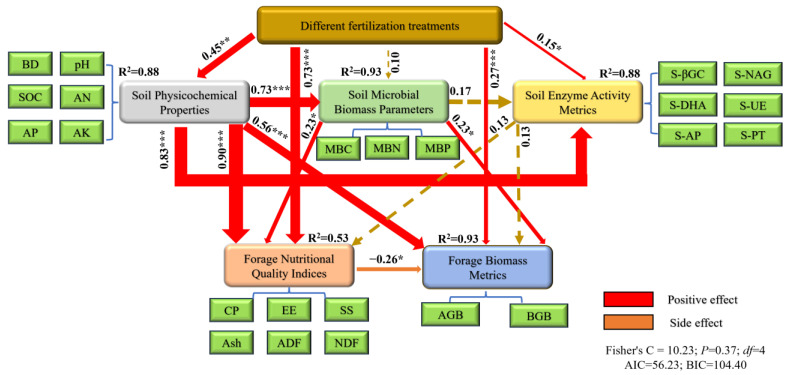
Structural equation model (SEM) analyzing the effects of different fertilization treatments on soil physicochemical properties, microbial biomass, enzyme activities, and forage yield/quality parameters. Significant path coefficients (*p* < 0.05) are indicated by solid lines, while non-significant relationships are shown as dashed lines. Red and orange arrows denote positive and negative correlations, respectively. Numbers adjacent to paths represent standardized coefficients (*β*), with arrow widths scaled proportionally to effect sizes. AGB: aboveground biomass; BGB: belowground biomass; pH: soil pH; BD: soil bulk density; SOC: soil organic carbon content; AN: alkali-hydrolyzable nitrogen content; AP: available phosphorus content; AK: available potassium content; MBC: soil microbial biomass carbon; MBN: soil microbial biomass nitrogen; MBP: soil microbial biomass phosphorus; CP: crude protein; EE: ether extract; SS: soluble sugars; Ash: crude ash; ADF: acid detergent fiber; NDF: neutral detergent fiber; S-βBG: soil β-glucosidase; S-DHA: soil dehydrogenase; S-NAG: soil N-acetyl-β-D-glucosaminidase; S-UE: soil urease; S-PT: soil protease; S-AP: soil acid phosphatase. Statistical significance levels: * *p* < 0.05, ** *p* < 0.01, *** *p* < 0.001.

**Figure 7 plants-14-03156-f007:**
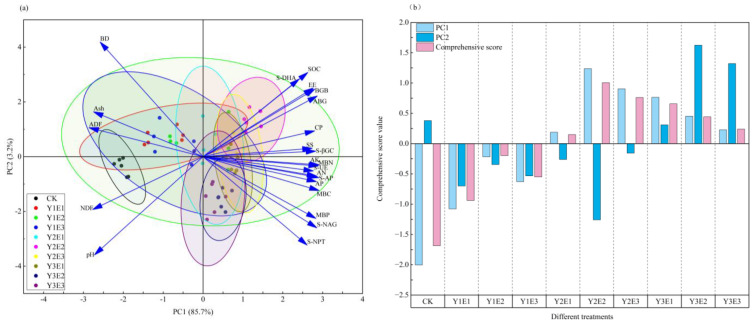
Comprehensive evaluation of soil physicochemical properties, microbial biomass, enzyme activities, and forage yield/quality parameters. Panel (**a**) displays principal component analysis (PCA) results, while panel (**b**) presents composite score evaluation. AGB: aboveground biomass; BGB: belowground biomass; pH: soil pH; BD: soil bulk density; SOC: soil organic carbon content; AN: alkali-hydrolyzable nitrogen content; AP: available phosphorus content; AK: available potassium content; MBC: soil microbial biomass carbon; MBN: soil microbial biomass nitrogen; MBP: soil microbial biomass phosphorus; CP: crude protein; EE: ether extract; SS: soluble sugars; Ash: crude ash; ADF: acid detergent fiber; NDF: neutral detergent fiber; S-βBG: soil β-glucosidase; S-DHA: soil dehydrogenase; S-NAG: soil N-acetyl-β-D-glucosaminidase; S-UE: soil urease; S-PT: soil protease; S-AP: soil acid phosphatase.

**Figure 8 plants-14-03156-f008:**
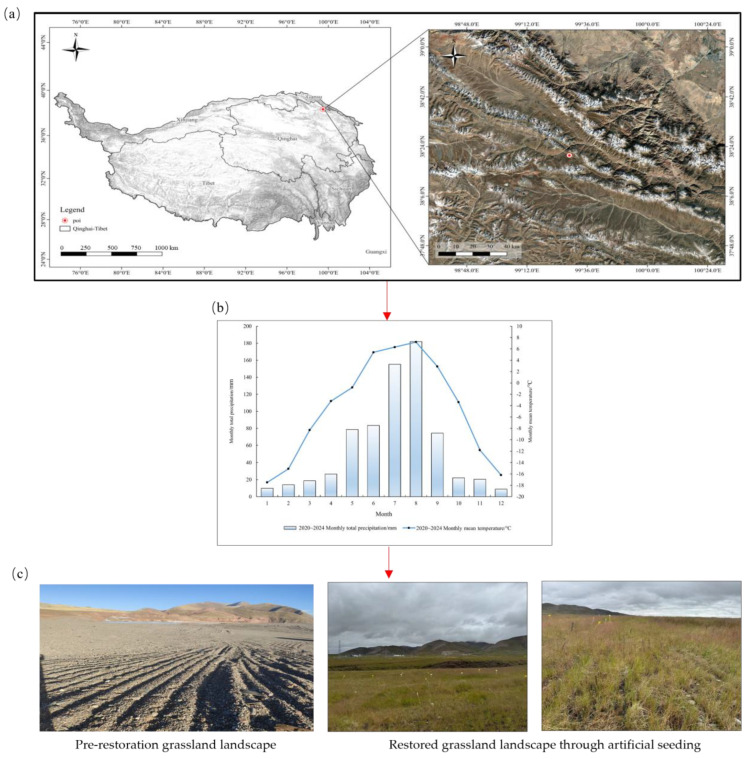
Geographic location, experimental layout, and field photographs of the study site. (**a**) Geographic location map of the experimental site; (**b**) Monthly mean temperature and total precipitation at the experimental site from 2020 to 2024 (meteorological data obtained from the Tianjun National Meteorological Station); (**c**) Field photograph of experimental plots.

**Table 1 plants-14-03156-t001:** Effects of different fertilization treatments on the physical and chemical properties of artificial grassland soil in 2023. In all figures and tables, lowercase letters (e.g., “a”, “b”) indicate statistically significant differences among treatments at *p* < 0.05. pH: soil pH; BD: soil bulk density; SOC: soil organic carbon content; AN: alkali-hydrolyzable nitrogen content; AP: available phosphorus content; AK: available potassium content.

Treatment	pH	BD/(g·cm^−1^)	SOC/(g·kg^−1^)	AN/(mg·kg^−1^)	AP/(mg·kg^−1^)	AK/(mg·kg^−1^)
CK	8.78 ± 0.04 a	1.74 ± 0.03 a	84.99 ± 3.56 g	171.66 ± 2.41 f	19.70 ± 0.34 f	238.72 ± 2.98 h
Y1E1	8.62 ± 0.04 b	1.70 ± 0.01 a	109.23 ± 3.51 f	194.12 ± 1.9 e	24.01 ± 0.87 e	270.09 ± 2.32 g
Y1E2	8.50 ± 0.03 c	1.47 ± 0.04 b	126.68 ± 1.34 de	212.99 ± 3.52 d	29.92 ± 1.13 d	297.25 ± 17.6 f
Y1E3	8.40 ± 0.04 d	1.18 ± 0.07 c	131.64 ± 1.76 cd	228.10 ± 4.9 c	30.62 ± 0.31 cd	361.57 ± 19.64 e
Y2E1	8.30 ± 0.04 e	1.13 ± 0.08 c	139.66 ± 1.10 ab	234.85 ± 2.44 bc	33.57 ± 0.67 bc	456.10 ± 10.11 bc
Y2E2	8.05 ± 0.03 g	1.08 ± 0.09 cd	146.80 ± 3.45 a	259.18 ± 2.08 a	37.47 ± 1.28 a	470.01 ± 1.86 a
Y2E3	8.20 ± 0.02 f	1.17 ± 0.13 c	138.63 ± 2.49 bc	252.68 ± 2.02 a	39.23 ± 0.29 a	461.60 ± 23.84 ab
Y3E1	8.30 ± 0.02 e	1.02 ± 0.06 cd	131.80 ± 2.71 bcd	236.34 ± 7.7 bc	36.38 ± 1.57 ab	453.61 ± 9.09 bc
Y3E2	8.41 ± 0.02 d	0.93 ± 0.01 d	125.17 ± 3.38 de	239.64 ± 1.49 b	36.98 ± 0.53 a	432.37 ± 13.69 cd
Y3E3	8.38 ± 0.02 de	1.05 ± 0.03 cd	123.37 ± 2.05 e	238.27 ± 1.10 bc	35.97 ± 2.45 ab	416.42 ± 32.75 d

**Table 2 plants-14-03156-t002:** Effects of different fertilization treatments on the physical and chemical properties of artificial grassland soil in 2024. In all figures and tables, lowercase letters (e.g., “a”, “b”) indicate statistically significant differences among treatments at *p* < 0.05. pH: soil pH; BD: soil bulk density; SOC: soil organic carbon content; AN: alkali-hydrolyzable nitrogen content; AP: available phosphorus content; AK: available potassium content.

Treatment	pH	BD/(g·cm^−1^)	SOC/(g·kg^−1^)	AN/(mg·kg^−1^)	AP/(mg·kg^−1^)	AK/(mg·kg^−1^)
CK	8.82 ± 0.04 a	1.64 ± 0.04 a	78.60 ± 2.52 f	180.01 ± 1.20 e	20.94 ± 0.41 e	247.81 ± 3.59 f
Y1E1	8.56 ± 0.06 b	1.50 ± 0.06 ab	116.40 ± 1.66 e	203.16 ± 2.53 d	24.35 ± 0.65 e	279.71 ± 7.84 e
Y1E2	8.40 ± 0.06 c	1.36 ± 0.07 b	131.32 ± 3.20 d	217.55 ± 1.12 d	31.82 ± 1.14 d	304.13 ± 16.62 e
Y1E3	8.30 ± 0.04 cd	1.12 ± 0.05 c	138.91 ± 4.00 cd	237.45 ± 7.01 c	32.89 ± 1.53 cd	390.91 ± 7.64 d
Y2E1	8.20 ± 0.04 de	1.06 ± 0.07 cd	150.22 ± 5.64 ab	240.68 ± 2.41 bc	40.23 ± 1.10 ab	462.10 ± 34.72 bc
Y2E2	7.84 ± 0.04 f	1.03 ± 0.07 cde	151.65 ± 2.94 a	257.37 ± 11.79 ab	41.73 ± 2.39 a	485.91 ± 6.06 a
Y2E3	8.11 ± 0.04 e	0.98 ± 0.08 cde	144.45 ± 4.23 abc	268.67 ± 4.91 a	39.89 ± 0.74 ab	470.29 ± 17.29 ab
Y3E1	8.17 ± 0.03 de	1.06 ± 0.09 cde	139.51 ± 5.12 bcd	246.51 ± 6.68 bc	38.29 ± 1.04 ab	463.12 ± 12.77 bc
Y3E2	8.36 ± 0.06 c	0.87 ± 0.02 e	135.53 ± 3.21 cd	248.16 ± 9.13 bc	37.63 ± 1.16 b	467.54 ± 35.64 abc
Y3E3	8.36 ± 0.03 c	0.90 ± 0.05 de	133.81 ± 3.11 cd	244.7 ± 3.96 bc	36.89 ± 2.24 bc	427.17 ± 34.97 c

**Table 3 plants-14-03156-t003:** Experimental design table of different fertilization treatments. Y: organic fertilizer, E: EM inoculant. The numbers 1~3 indicate different application rates.

Treatment Group	Organic Fertilizer (t·hm^−2^)	Effective Microorganisms (t·hm^−2^)
CK	10.00	—
Y1E1	10.00	0.45
Y1E2	10.00	0.60
Y1E3	10.00	0.75
Y2E1	20.00	0.45
Y2E2	20.00	0.60
Y2E3	20.00	0.75
Y3E1	40.00	0.45
Y3E2	40.00	0.60
Y3E3	40.00	0.75

## Data Availability

The original contributions presented in this study are included in this article; further inquiries can be directed to the corresponding authors.

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
