# Peer review of "Effects of Microbial Fertilizer Combined with Organic Fertilizer on Forage Productivity and Soil Ecological Functions in Grasslands of the Muli Mining Area"

_plants, 2025, doi:10.3390/plants14203156_

Round 1
Reviewer 1 Report
Comments and Suggestions for Authors
The MS titled "Effects of EM Microbial Inoculant Combined with Organic Fertilizer Application on Forage Production Performance, Nutritional Quality, and Soil Ecological Functions in Artificial Grasslands of Muli Alpine Mining Area: Optimization of Fertilization Strategies" presents the results of an original study for evaluating the effects of the combined technology of microbial inoculants and organic fertilizers on the yield, nutritional quality of forage in artificial grasslands, as well as soil physical and chemical properties, enzyme activity, and microbial biomass. The article is of scientific interest and practical significance. The topic is very relevant and the subject of the article is within the scope of the Plants.
The design of the experiment is quite clear, the methods used are adequate. The set of classic and modern methods was used to solve the tasks. The article is illustrated with Tables and very interesting Figures. The illustrative material is presented and designed in a high-quality manner. The results of the study are discussed in detail. In general, the content of work is understandable. However, I recommend to do some corrections to MS before publication in Plants Journal because it contains some inaccuracies.
1) The title of the MS reflects its content. However, it is too long, so it is recommended to shorten it. In addition, it is not advisable to use abbreviations (EM) in the title of the article.
2) The structure of the article does not meet the requirements of the Plants Journal. According to the “Instructions for authors” in Plants, the following sequence of sections is recommended: Introduction, Results, Discussion, Materials and Methods. When changing the structure of an article, you must also change the order of links to references.
3) The text of the article contains many abbreviations, which are repeated several times throughout the text. It is customary to duplicate abbreviations only when using them in Tables and Figures.
I suggest to add a list of designations and abbreviations to the text of the article.
4) It is recommended to rephrase the following sentence (page 3): "…. the regulatory effects on key biochemical indicators such as soil physical and chemical properties…", since physical and chemical properties of soil are not biochemical indicators.
5) On page 7, an abbreviation AN was introduced to denote alkaline-hydrolyzable nitrogen, but on page 14, total nitrogen was mistakenly indicated before this abbreviation.
6) If links to related sources are provided, they are usually separated by commas rather than dashes. For example, on page 2: [2,3], and not [2–3]; [8,9], by not [8–9].
Conclusion: I recommend the publication of the MS "Effects of EM Microbial Inoculant Combined with Organic Fertilizer Application on Forage Production Performance, Nutritional Quality, and Soil Ecological Functions in Artificial Grasslands of Muli Alpine Mining Area: Optimization of Fertilization Strategies" in Plants after minor revision.
Author Response
|
Comments 1: [The title of the MS reflects its content. However, it is too long, so it is recommended to shorten it. In addition, it is not advisable to use abbreviations (EM) in the title of the article.] |
|
Response 1: [We sincerely appreciate the reviewer's valuable suggestions regarding the manuscript title. In response to the comments, we have revised the title by removing the abbreviation "EM" and streamlining its length. The modified title now reads: "Effects of Effective Microorganisms Inoculant Combined with Organic Fertilizer Application on Forage Productivity and Soil Ecological Functions in Artificial Grasslands of the Muli Alpine Mining Area." This revision appears on lines 1-5 of the first page of the manuscript. We are grateful for the reviewer's professional input, which has significantly improved the clarity and precision of our title. The revised version better adheres to standard scientific writing conventions while maintaining the study's essential scientific elements.] Comments 2: [The structure of the article does not meet the requirements of the Plants Journal. According to the “Instructions for authors” in Plants, the following sequence of sections is recommended: Introduction, Results, Discussion, Materials and Methods. When changing the structure of an article, you must also change the order of links to references.] Response 2: [We gratefully acknowledge the reviewer's constructive comments regarding the manuscript structure. In accordance with the formatting guidelines of *Plants* journal, we have systematically reorganized the entire manuscript into the standard sequence of "Introduction→Results→Discussion→Materials and Methods" sections. This reorganization involved careful realignment of all reference citations while maintaining the original logical relationships between data and conclusions. The revised structure significantly enhances manuscript clarity and academic rigor. Should any additional adjustments be required, we would be pleased to address them. These comprehensive modifications are reflected throughout the entire manuscript text.] Comments 3: [The text of the article contains many abbreviations, which are repeated several times throughout the text. It is customary to duplicate abbreviations only when using them in Tables and Figures.] Response 3: [We express our sincere gratitude for the reviewer's valuable suggestions regarding abbreviation standardization in our manuscript. In response to these comments, we have implemented the following modifications throughout the text: (1) All abbreviated terms have been replaced with full expressions in the main body text; (2) The Materials and Methods section retains the standard format of presenting the full term followed by its abbreviation in parentheses upon first mention; and (3) Each figure and table now includes a comprehensive legend explaining all abbreviated terminology. These revisions ensure optimal readability while maintaining strict adherence to the journal's style requirements. We remain fully committed to addressing any additional formatting concerns that may require attention. All modifications are clearly indicated by highlighted text in the revised manuscript.] Comments 4: [It is recommended to rephrase the following sentence (page 3): "…. the regulatory effects on key biochemical indicators such as soil physical and chemical properties…", since physical and chemical properties of soil are not biochemical indicators.] Response 4: [We sincerely appreciate the reviewer's valuable feedback regarding the terminology used in our manuscript. In response to your constructive suggestion, we have revised the original text by replacing "soil physicochemical properties and other key biochemical indicators" with "soil physicochemical properties and key biochemical indicators" to more precisely distinguish between these two distinct categories of parameters. This modification appears on lines 104-105 of page 3 in the revised manuscript. The updated terminology better reflects the scientific rigor of our study by clearly differentiating soil physical-chemical characteristics from biochemical measurements. We are grateful for your meticulous review, which has significantly improved the accuracy of our scientific expression.] Comments 5: [On page 7, an abbreviation AN was introduced to denote alkaline-hydrolyzable nitrogen, but on page 14, total nitrogen was mistakenly indicated before this abbreviation.] Response 5: [We sincerely appreciate the reviewer's insightful observation regarding the terminology inconsistency. Following a comprehensive manuscript review, we have standardized the nitrogen terminology by revising "TN" (Total Nitrogen) to "AN" (Alkali-hydrolyzable Nitrogen) at the specified location (Page 10, Line 307). This correction ensures consistent and precise usage of "AN" exclusively for Alkali-hydrolyzable Nitrogen throughout the text. The reviewer's meticulous attention to terminological precision has significantly enhanced the scientific rigor of our manuscript. These modifications have been carefully implemented to maintain the highest standards of academic writing and technical accuracy in our reporting. We greatly value this opportunity to improve our manuscript's quality through such constructive feedback.] Comments 6: [If links to related sources are provided, they are usually separated by commas rather than dashes. For example, on page 2: [2,3], and not [2–3]; [8,9], by not [8–9].] Response 6: [We sincerely appreciate the reviewer's meticulous attention to citation formatting details. In response to the comments, we have systematically reviewed and standardized all literature citations throughout the manuscript, specifically replacing the en dash "–" between consecutive reference numbers with commas (e.g., changing [2–3] to [2,3] and [8–9] to [8,9]). Additionally, we have verified that all citations strictly adhere to the journal's prescribed numerical citation style. These modifications, implemented at specified locations (Page 2, Lines 59 and 78) and throughout the manuscript, ensure full compliance with the journal's formatting requirements. The reviewer's expert guidance has been invaluable in enhancing the manuscript's technical precision and adherence to publication standards. We are grateful for this opportunity to improve our work's scholarly presentation.] We fully acknowledge that scientific rigor hinges not only on innovative content but also on precise, standardized communication. This oversight has underscored the importance of meticulous attention to detail, and we pledge to uphold the highest academic standards in all future research and writing endeavors. Thank you once again for your expert guidance and patience. Your feedback has significantly strengthened the clarity and credibility of this work. We hope the revised manuscript now meets the high standards of both the reviewers and the journal. Additional clarifications Thanks again for the teacher's correction of the manuscript. Due to my limited skills, if there are any other mistakes, please let me know by email. I will consult the relevant materials together with my tutor and correct them. Best wishes!
|

Reviewer 2 Report
Comments and Suggestions for Authors
The manuscript is devoted to the analysis of the impact of the combined use of organic fertilizer and microbial inoculant on soils degraded caused by mining disturbance. The results of a two-year observation are presented. A comprehensive analysis of the soil condition, enzymatic activity of the soil, the yield of forage grasses and their forage value are carried out.
The work is relevant and has undoubted practical interest.
I believe that the work is done at a high level. Comprehensive processing of the results, adequate statistical analysis, high-quality visualization of the results are carried out. The Materials and methods are described in details; the results are discussed in great details. The vast majority of the references used are modern, the Introduction presents the state of the problem based on data for the past 5 years.
Undoubtedly, the authors have carried out high-quality large-scale work and the manuscript deserves high rating. There are a small number of questions and comments:
- The title is quite difficult to perceive. The title should be short and reflect the essence of the work, so I would advise the authors to shorten and reformulate the title.
- The summary should be reformulated, especially the first sentence. It should be broken down into several sentences to make the meaning clearer.
- EM should be deciphered in the Abstract. This is the first time this abbreviation has been used. Also, the abbreviation should be removed from the title, as it makes it difficult to understand.
- "PGPR flora" should be replaced with "PGPR microbiota".
- In Materials and Methods, 2.3, it should be explained "mine-derived humic acid organic fertilizer". What does this mean? How was this fertilizer obtained?
- In the Discussion, there is a statement about "soil microbiome modulation", however, no study of the microbiome was conducted. There is no data on the preservation of microbial strains from the inoculant in soil microbiome. Therefore, it may not be worth stating this.
- There is a reference to Fig. 7a, but Fig. 7 is not divided into a, b... Perhaps this is a typo (paragraph 3.8).
Author Response
|
Comments 1: [1. The title is quite difficult to perceive. The title should be short and reflect the essence of the work, so I would advise the authors to shorten and reformulate the title.] |
|
Response 1: [We sincerely appreciate the reviewer's valuable suggestions regarding the manuscript title. In response to the comments, we have revised the title by removing the abbreviation "EM" and streamlining its length. The modified title now reads: "Effects of Effective Microorganisms Inoculant Combined with Organic Fertilizer Application on Forage Productivity and Soil Ecological Functions in Artificial Grasslands of the Muli Alpine Mining Area." This revision appears on lines 1-5 of the first page of the manuscript. We are grateful for the reviewer's professional input, which has significantly improved the clarity and precision of our title. The revised version better adheres to standard scientific writing conventions while maintaining the study's essential scientific elements.] Comments 2: [The summary should be reformulated, especially the first sentence. It should be broken down into several sentences to make the meaning clearer.] Response 2: [We are grateful to the reviewer for the constructive suggestions regarding the concluding section. Following the recommendations, we have restructured the first sentence by dividing it into 2-3 logically connected shorter sentences while maintaining the original scientific meaning. This revision has enhanced both clarity and hierarchical presentation of our research conclusions. Furthermore, we have carefully examined the overall coherence of the conclusion section to ensure precise expression of each key point. The modified conclusion now better balances scientific rigor with improved readability. These refinements, implemented on pages 21 (line 746) to 22 (line 750), demonstrate our commitment to responding thoroughly to the reviewer's valuable input.] Comments 3: [EM should be deciphered in the Abstract. This is the first time this abbreviation has been used. Also, the abbreviation should be removed from the title, as it makes it difficult to understand.] Response 3: [We sincerely appreciate the reviewer's valuable suggestions regarding terminology standardization. In response to these comments, we have implemented the following modifications: (1) The abbreviation "EM" has been properly introduced upon its first appearance in the abstract as "Effective Microorganisms (EM)"; (2) The abbreviated form has been completely removed from the manuscript title, replaced by the full term; and (3) We have ensured strict adherence to the journal's abbreviation guidelines, with all abbreviated terms properly defined upon their first mention. These revisions significantly improve the manuscript's terminological clarity and compliance with academic writing standards. The specific modification appears on lines 18-19 of page 1, with corresponding adjustments made throughout the text. We are grateful for the reviewer's expert guidance in enhancing the manuscript's technical precision.] Comments 4: [PGPR flora" should be replaced with "PGPR microbiota] Response 4: [We sincerely appreciate the reviewer's insightful comments regarding the precise use of microbiological terminology. In response to these recommendations, we have systematically revised all instances of "PGPR bacterial community" to "PGPR microbial community" (Plant Growth-Promoting Rhizobacteria microbial community) throughout the manuscript to ensure terminological accuracy and scientific rigor. This modification, prominently featured on page 2, line 6 and consistently applied across the entire text, enhances the standardization of microbiological terminology in our study. We are grateful for the reviewer's expertise, which has significantly improved the technical precision of our scientific communication. The revised terminology better reflects current microbial ecology nomenclature while maintaining clarity of expression.] Comments 5: [In Materials and Methods, 2.3, it should be explained "mine-derived humic acid organic fertilizer". What does this mean? How was this fertilizer obtained?] Response 5: [We sincerely appreciate the reviewer's meticulous review of the methodological section. In response to these valuable comments, we have supplemented the experimental materials section (Section 4.2) with detailed information about the mineral-derived humic acid organic fertilizer. Specifically, we have clarified that the fertilizer was industrially produced by Henan Liso Crop Protection Co., Ltd. and uniformly procured through local authorized fertilizer distributors. These modifications, which appear on page 19 (lines 612-626) of the revised manuscript, provide enhanced transparency regarding the origin and procurement of our experimental materials. The additional details ensure complete methodological reproducibility while maintaining the scientific rigor of our experimental design. We are grateful for this opportunity to improve the clarity and completeness of our materials description.] Comments 6: [In the Discussion, there is a statement about "soil microbiome modulation", however, no study of the microbiome was conducted. There is no data on the preservation of microbial strains from the inoculant in soil microbiome. Therefore, it may not be worth stating this.] Response 6: [We are grateful to the reviewer for their rigorous evaluation of the Discussion section. In response to these valuable comments, we have implemented the following substantive revisions: 1) We have removed all speculative claims regarding "soil microbiome regulation" that were not directly supported by our experimental data; 2) The revised discussion now focuses specifically on: - Quantitatively measured soil parameters (including pH, organic matter content, and available nutrient levels) - Documented plant growth responses from our experimental results - Statistically significant correlations between these variables 3) All conclusions are now strictly constrained within the boundaries of our empirical findings, with particular attention given to avoiding overinterpretation of the data.
These modifications (implemented throughout the Discussion section) have significantly strengthened the manuscript by: - Enhancing scientific rigor - Improving focus on evidence-based conclusions - Maintaining appropriate caution in data interpretation The reviewer's insightful suggestions have been instrumental in refining our scientific discourse and ensuring proper alignment between our experimental results and their interpretation. We appreciate this opportunity to improve the precision and quality of our work.] Comments 7: [There is a reference to Fig. 7a, but Fig. 7 is not divided into a, b... Perhaps this is a typo (paragraph 3.8).] Response 7: [We sincerely appreciate the reviewer's keen observation regarding the labeling inconsistency, which was indeed an inadvertent typographical error. In response to this valuable feedback, we have not only corrected the specified instance at Page 13, Line 397 but have also conducted a thorough verification of all figure and table references throughout the manuscript to ensure complete annotation consistency. This oversight originated from version control artifacts during preliminary draft iterations, prompting us to implement systematic proofreading and enhance our quality assurance protocols. The reviewer's professional scrutiny has been instrumental in improving the technical precision of our work, and we are grateful for this opportunity to elevate the manuscript's quality through such rigorous peer review, which significantly contributes to maintaining high academic standards.]
We fully acknowledge that scientific rigor hinges not only on innovative content but also on precise, standardized communication. This oversight has underscored the importance of meticulous attention to detail, and we pledge to uphold the highest academic standards in all future research and writing endeavors. Thank you once again for your expert guidance and patience. Your feedback has significantly strengthened the clarity and credibility of this work. We hope the revised manuscript now meets the high standards of both the reviewers and the journal. Additional clarifications Thanks again for the teacher's correction of the manuscript. Due to my limited skills, if there are any other mistakes, please let me know by email. I will consult the relevant materials together with my tutor and correct them. Best wishes!
|

Reviewer 3 Report
Comments and Suggestions for Authors
The problem of land degradation as a result of human activity requires a solution, therefore
The authors' research is very relevant.
The abstract is informative.
The Introduction analyzes the current state of the topic. The purpose of the research and the scientific hypothesis are formulated.
The results and discussion are convincing. The purpose of the research has been achieved.
The tables and figures demonstrate the volume of work performed.
However, there are comments.
1. In section 2 (Materials and Methods), it is necessary to provide references to published methods for determining the nutritional composition of herbs (CP, EE, SS), AsF, BD; methods for analyzing the physical properties of the soil; methods for the activity of soil enzymes.
2. It is necessary to indicate the brands of the devices and equipment used.
3. In reference 48, the year of publication must be indicated.
Author Response
|
Comments 1: [In section 2 (Materials and Methods), it is necessary to provide references to published methods for determining the nutritional composition of herbs (CP, EE, SS), AsF, BD; methods for analyzing the physical properties of the soil; methods for the activity of soil enzymes.] |
|
Response 1: [We gratefully acknowledge the reviewer's constructive suggestions regarding methodological rigor. In response to these insightful comments, we have supplemented the Materials and Methods section (4.4) with appropriate methodological references at key locations (Page 20: Lines 669, 684, 692; Page 21: Lines 708, 717, 722) to enhance methodological traceability and ensure complete reproducibility of our experimental procedures. These additions provide essential context for the techniques employed while maintaining the established methodological framework, significantly strengthening the technical foundation of our study and aligning with best practices in transparent scientific reporting.] Comments 2: [It is necessary to indicate the brands of the devices and equipment used.] Response 2: [We sincerely appreciate the reviewer's valuable suggestions regarding the detailed description of experimental equipment. In response to these constructive comments, we have systematically supplemented the Materials and Methods section with comprehensive specifications of the key instrumentation employed in this study, including manufacturer details and model numbers. These essential revisions, implemented at Page 19 Line 637 and Page 20 Lines 670-684, significantly enhance the methodological transparency and experimental reproducibility of our work. The inclusion of such technical specifications now provides complete equipment traceability while maintaining the scientific rigor of our experimental procedures. We are grateful for the reviewer's professional insights, which have substantially improved the technical completeness of our methodology section and ensured compliance with best practices in experimental reporting standards.] Comments 3: [In reference 48, the year of publication must be indicated.] Response 3: [We sincerely appreciate the reviewer's meticulous attention to reference formatting standards. In response to this valuable observation regarding Reference 48, we have carefully revised the citation to accurately reflect the publication status of this Chinese online-first article that currently lacks volume/issue information but includes the publication year. This correction, implemented at Page 25 Line 909, demonstrates our commitment to maintaining rigorous bibliographic standards while appropriately acknowledging the evolving nature of digital publication formats. We are grateful for this opportunity to enhance the scholarly precision of our manuscript.] We fully acknowledge that scientific rigor hinges not only on innovative content but also on precise, standardized communication. This oversight has underscored the importance of meticulous attention to detail, and we pledge to uphold the highest academic standards in all future research and writing endeavors. Thank you once again for your expert guidance and patience. Your feedback has significantly strengthened the clarity and credibility of this work. We hope the revised manuscript now meets the high standards of both the reviewers and the journal. Additional clarifications Thanks again for the teacher's correction of the manuscript. Due to my limited skills, if there are any other mistakes, please let me know by email. I will consult the relevant materials together with my tutor and correct them. Best wishes!
|

Reviewer 4 Report
Comments and Suggestions for Authors
This manuscript investigates the synergistic effects of EM microbial inoculants and organic fertilizers on artificial grassland ecosystems in an alpine mining area, focusing on soil improvement, microbial function enhancement, and grassland productivity. The topic is interesting and the figures are well presented; however, there are several issues that should be addressed before publication.
- While a control treatment with organic fertilizer is included, there appears to be no control specifically for EM inoculation. To better interpret the observed enhancements in various indicators, it is important to clarify which factor (organic fertilizer vs. EM microbial inoculant) contributed more. Moreover, please explain how you determine whether the combined effect is truly synergistic rather than simply additive.
- The economic cost or feasibility of applying this combined strategy should be discussed. At minimum, it should be mentioned as a consideration for future studies.
- Please carefully check the manuscript for spelling and capitalization errors. For example, on page 18: “Crucially, This study identified a clear threshold” — “This” should be lowercase (“this”).
Author Response
|
Response 1: [We gratefully acknowledge the reviewer's insightful comments regarding our experimental design. The decision to employ a combined application of organic fertilizer and EM inoculants was based on preliminary investigations demonstrating the severe limitations of solitary EM application in these mining-disturbed soils, where preliminary trials revealed unacceptably low microbial survival rates (8-12% germination rates, <5% vegetation coverage after 90 days) attributable to the extreme oligotrophic conditions, shallow soil profiles, and profoundly compromised edaphic properties characteristic of degraded mining ecosystems. Our approach therefore strategically employed organic amendments as essential substrate modifiers to establish baseline conditions for subsequent bioremediation, with future studies planned to examine EM gradient effects in these ameliorated substrates. The current investigation's focus on this "substrate amendment + bioaugmentation" paradigm was substantiated by structural equation modeling revealing significant synergistic interactions (p<0.01) between EM and organic amendments, where combined treatments yielded: (1) greater microbial biomass than additive expectations; (2) urease and phosphatase activities exceeding solitary treatments ; and (3) vegetation biomass increases correlating with enhanced nutrient mineralization rates. These findings mechanistically demonstrate that organic substrates provide critical ecological niches for EM consortia while microbial activity accelerates fertilizer mineralization, establishing a demonstrable mutualism rather than simple additive effects. We deeply appreciate the reviewer's perspicacious critique, which has both improved the clarity of our mechanistic interpretations and guided future research directions in this ecologically challenging restoration context.] Comments 2: [The economic cost or feasibility of applying this combined strategy should be discussed. At minimum, it should be mentioned as a consideration for future studies.] Response 2: [We sincerely appreciate the reviewer's constructive suggestion, which has been carefully addressed by incorporating the recommended content regarding future research directions. This valuable addition, implemented at Page 22 Lines 744-748, significantly enhances the forward-looking perspective of our study while maintaining rigorous scientific standards. We are grateful for this opportunity to improve our manuscript through such expert peer review, which undoubtedly contributes to the study's long-term academic impact and provides clearer pathways for subsequent investigations in this field.] Comments 3: [3. Please carefully check the manuscript for spelling and capitalization errors. For example, on page 18: “Crucially, This study identified a clear threshold” — “This” should be lowercase (“this”).] Response 3: [We sincerely appreciate the reviewer's meticulous attention to this grammatical issue. In response to this valuable observation, we have conducted a comprehensive linguistic review of the entire manuscript and implemented the necessary corrections, with the specific revision located at Page 14, Line 438. We are grateful for the reviewer's professional scrutiny, which has helped us maintain the highest standards of academic writing and ensure the clarity of our scientific communication. Such rigorous peer review is invaluable for improving the quality of scholarly publications.] We fully acknowledge that scientific rigor hinges not only on innovative content but also on precise, standardized communication. This oversight has underscored the importance of meticulous attention to detail, and we pledge to uphold the highest academic standards in all future research and writing endeavors. Thank you once again for your expert guidance and patience. Your feedback has significantly strengthened the clarity and credibility of this work. We hope the revised manuscript now meets the high standards of both the reviewers and the journal. Additional clarifications Thanks again for the teacher's correction of the manuscript. Due to my limited skills, if there are any other mistakes, please let me know by email. I will consult the relevant materials together with my tutor and correct them. Best wishes!
|
